# Comparison of Phenolic and Flavonoid Compound Profiles and Antioxidant and α-Glucosidase Inhibition Properties of Cultivated Soybean (*Glycine max*) and Wild Soybean (*Glycine soja*)

**DOI:** 10.3390/plants10040813

**Published:** 2021-04-20

**Authors:** Qianru Chen, Xianxian Wang, Xiaolong Yuan, John Shi, Chengsheng Zhang, Ning Yan, Changliang Jing

**Affiliations:** 1Tobacco Research Institute of Chinese Academy of Agricultural Sciences, Qingdao 266101, China; ycscqr@163.com (Q.C.); yuanxiaolong@caas.cn (X.Y.); zhangchengsheng@caas.cn (C.Z.); 2College of Plant Health and Medicine Scientific Research Center, Qingdao Agricultural University, Qingdao 266101, China; pretty0601@163.com; 3Guelph Food Research Centre, Agriculture and Agri-Food Canada, Guelph, ON N1G 5C9, Canada; john.shi@agr.gc.ca

**Keywords:** cultivated soybean (*Glycine max*), wild soybean (*Glycine soja*), antioxidant activities, metabolomics, phenolic compounds

## Abstract

Wild soybean (*Glycine soja* Sieb.et *Zucc;* WS) has been used as a traditional food in China for many years and contains significantly higher levels of isoflavones than cultivated soybean (*Glycine max*; CS), but the secondary metabolites, including flavonoids and the phenolic composition differences between them, remain unclear. The results showed that WS possessed significantly higher total phenolic and flavonoid content and exhibited better antioxidant and α-glucosidase inhibition activities as well as excellent protective effects against H_2_O_2_-induced oxidative injury in a human endothelial cell line. Through metabolomic analysis, 642 metabolites were identified, and 238 showed differential expression, with 151 upregulated and 87 downregulated. A total of 79 flavonoid compounds were identified, 42 of which were upregulated in WS. 2′-Hydroxygenistein, garbanzol, protocatechuic aldehyde, ligustilide, and resveratrol were the most discriminated compounds in WS. The metabolic pathway analysis of differential metabolites related to the biosynthesis of flavonoids and phenolic acids were the biosynthesis of phenylpropanoids, flavonoids, isoflavonoids, flavones, and flavonols. This study substantially elucidated differences in the content of flavonoids and biological activities between WS and CS, which is useful information for the effective utilization of these two black soybean species in food processing.

## 1. Introduction

Soybean is widely consumed as a nutritionally rich food throughout the world. It is composed of proteins, oils, carbohydrates, and dietary fiber, as well as significant amounts of vitamins, minerals, and phytochemicals [1]. China is very rich in leguminous plants, and there are more than 20 types of legumes with different seed colors, including yellow, brown, red, green, and black [2]. Compared to other legumes, black soybean contains higher amounts of antioxidants such as phenolic acids, flavonoids, anthocyanins, and proanthocyanidins, which have various biological activities [3]. Consequently, black soybean has been widely used in traditional oriental medicines and has attracted more attention in China.

WS has a small black seed coat and can be used as a traditional food. It is widely distributed in China and exhibits great adaptability to various environments. In our previous studies, the results confirmed the positive health impacts of WS, such as improvements in the blood lipid profile and reductions in hepatic steatosis and adipocyte size in highly fed mice [4]. A number of studies have verified that the health benefits associated with polyphenols in soybean include the effects from phenolic acids, flavonoids, and anthocyanins [5]. Moreover, these compounds are considered the most effective antioxidants in the human diet. Notably, previous studies reported 68 differential metabolites of CS and WS under different stresses and compared the metabolic profiling of CS and WS and detected 45 primary and 3 secondary metabolites [6,7]. Investigation of their phytochemicals and biological activities is necessary to identify soybean varieties with large health benefits.

Metabolomic technology has been widely used in research on traditional Chinese medicine, clinical disease diagnosis, toxicology, and other scientific fields [8]. Plant metabolomics mainly involves the study of corresponding changes in small molecule metabolites of different species and different gene types or ecological types during different growth periods, including changes in species and quantity and their change rules [9]. In recent years, applications of untargeted metabolomics for food authentication have gained increasing interest, and many studies have been conducted using analytical tools for comprehensive analyses of phytochemicals in fruit, vegetables, and grain [10].

WS contains significantly higher levels of isoflavones than CS [11]. Based on our knowledge so far, no studies have compared CS and WS in terms of total phenolic content (TPC), total flavonoid content (TFC), or antioxidant and α-glucosidase inhibition properties. Furthermore, no comparative information on the use of metabolomics as a tool to evaluate the specific flavonoids and polyphenol composition in WS or their differences in biological activities is available. Therefore, in the present study, we aimed to evaluate the types and relative content of secondary metabolites including flavonoids and phenolics in WS using ultra-high performance liquid chromatography coupled with triple quadrupole mass spectrometry in a metabolomic approach. The metabolic pathway and hierarchical cluster analyses were also undertaken to identify flavonoid compounds. The insights gained in this study expand the current knowledge of WS and may facilitate further research and utilization of this species.

## 2. Materials and Methods

### 2.1. Samples

WS and CS were selected as experimental black soybean varieties, and seeds of the two varieties were collected. One kilogram of WS seed samples was collected from Kenli District, Dongying City, in the Yellow River Delta (37°39′43.58″ N, 118°40′48.19″ E), which is a typical distribution area of WS in China. One kilogram of CS seed samples was collected from the same place they were cultivated. The difference in appearance between the species is mainly in the size of the small black seeds that WS produces, whereas CS produces large black seeds (Appendix A). The samples were dried to a constant weight using a freeze-dryer (Alpha 1–2 LD Plus; Christ, Osterode am Harz, Germany), and were then stored at −20 °C. For all assays, triplicates were analyzed for each WS and CS sample.

### 2.2. Chemicals

All chemicals used in the study were of chromatographic grade. Methanol, acetonitrile, and chromatographic ethanol were purchased from Merck (Darmstadt, Germany). Double-deionized ultrapure water, prepared using the Milli-Q system (Millipore, Bedford, MA, USA), was used. α-Glucosidase (type I, from *Saccharomyces cerevisiae*) was purchased from Sigma Aldrich (St. Louis, MO, USA).

### 2.3. Extraction of Phenolic Compounds

Free and bound phenolics of WS and CS samples were extracted according to the method of Yu et al. [12]. Seeds of the two varieties stored at −20 °C were vacuum freeze-dried. The dried samples were ground with a grinder (MM 400, Retsch, Haan, Germany) at 30 Hz for 1.5 min, and 100 mg of each powdered sample were dissolved in a 1.0 mL solution (70% methanol). The resulting mixtures were stored at 4 °C overnight and vortexed three times to ensure complete extraction. After filtration through a filter membrane (0.22 μm pore size), we obtained the free and bound phenolic extracts. Equal amounts (1 mL) of the free and bound phenolics extracts were pipetted into a 2.5 mL centrifuge tube and then homogenized in a vortex to obtain a solution containing the total phenolic compounds. This solution was stored in a refrigerator at 4 °C for two days until analysis of the phenolic compounds and biological activity.

### 2.4. Determination of Total Phenolic and Total Flavonoid Content

Briefly, 1 mL of sample extract (25 mg/mL) was combined with 0.5 mL of Folin-Ciocalteu reagent, to which 1.5 mL of sodium carbonate solution (Na_2_CO_3_, 20%, *w/v*) were added. The mixture was then incubated for 2 h in the dark at 40 °C. The absorbance was recorded at 760 nm using ethanol as the blank. A standard curve was prepared using gallic acid (0.1–0.8 mg/mL), and TPC was expressed as gallic acid equivalents (GAE; mg GAE/g extracts).

The sample extract was mixed with 0.3 mL of NaNO_2_ (5%), 0.3 mL of AlCl_3_ (10%) was added after 5 min, and 2 mL of NaOH (1 M) and water were then added to obtain a final volume of 10 mL. The absorbance was measured at 510 nm. A standard curve was prepared using quercetin, and TFC was expressed as quercetin equivalents (QE; mg QE/g extracts).

### 2.5. Antioxidant Assays

#### 2.5.1. 1,1-Diphenyl-2-picrylhydrazyl (DPPH) Radical-Scavenging Activity Assay

The DPPH radical scavenging activity of the soybean extracts was determined using a method described by Jing et al. [13] with some modifications. In brief, extracts were dissolved in a 70% ethanol solution to prepare various concentrations, and 750 µL of a 0.1 mM ethanol solution of DPPH was mixed with various sample solutions (0–1 mg/mL). After incubation at room temperature for 30 min in the dark, absorbance was measured at 517 nm. Ethanol was used as the blank control, and ascorbic acid was used as the positive control.

The DPPH scavenging activity was evaluated according to the following equation:DPPH radical scavenging activity (%) = [1 − (A_1_ − A_2_)]/A_0_] × 100%(1)
where A_0_ is the absorbance of the ethanol solution by DPPH without extracts, A_1_ represents the absorbance of the ethanol solution by DPPH with tested extracts, and A_2_ represents the absorbance of the ethanol by tested extracts.

#### 2.5.2. 2,2′-Azino-bis(3-ethylbenzothiazoline-6-sulfonic Acid) (ABTS) Radical-Scavenging Activity Assay

The ABTS radical scavenging activity of the soybean extracts was determined using the method described previously with some modifications [13]. The stock solution of the ABTS solution was prepared by reacting a 7.4 mM ABTS solution with 3.8 mM potassium persulfate and was then kept in the dark at ambient temperature for 16 h. For the assay, the ABTS solution was diluted with ethanol to an absorbance value of 0.7 ± 0.02 at 734 nm. The diluted ABTS solution (3.9 mL) was added to 100 μL samples (0–1 mg/mL), absorbance was measured at 734 nm after 30 min, and the ABTS radical scavenging activity was calculated using the same equation as that for the DPPH scavenging activity.

### 2.6. α-Glucosidase Inhibitor Activity Assay

α-Glucosidase inhibitor activity was determined according to the method described by Yuan et al. [14]. A α-glucosidase solution (3.92 U/mL) and a *p*-nitrophenyl-α-D-glucopyranoside solution (PNPG: 6 mM) were prepared by 100 mM phosphate buffer saline. Briefly, a 100 μL sample solution (10 mg/mL extract in 3% dimethyl sulfoxide (DMSO)) was mixed with 10 μL of α-glucosidase and incubated at 37 °C for 10 min. Afterward, 200 μL of PNPG was added and incubated for another 20 min. The reaction was terminated by adding 1 mL of Na_2_CO_3_ and further diluted with 4 mL of distilled water. The absorbance was measured at 400 nm. A solution with 3% DMSO was used as the control, and acarbose was used as the positive control. The α-glucosidase inhibitor rate was calculated as follows:α-Glucosidase inhibitor rate (%) = [1 − (A_1_ − A_0_/A_2_ − A_0_)] × 100%(2)
where A_0_ represents the absorbance of the background solution with α-glucosidase, A_1_ represents the absorbance of the samples, and A_2_ represents the absorbance of the control without sample.

### 2.7. Cell Treatment and Cell Viability Assay

Human umbilical vein endothelial cells (HUVECs) (1.5 × 10^5^ cells/well) were plated in 24-well plates. After stimulation with H_2_O_2_ and/or pre-incubation with 200 μg/mL WS extracts or CS extracts, the cells were cultured for 24 h. Next, the supernatant was aspirated, and the cells were fixed with 100 μL of 10% cold trichloroacetic acid at 4 °C for 1 h, washed with distilled water five times, and air-dried. One hundred microliters of 4 mg/mL sulforhodamine B solution (Sigma Aldrich, Carlsbad, CA, USA) were added into each well and dyed for 15 min at room temperature, followed by removal of the supernatant, washing five times with 1% acetic acid, and air-drying. Afterward, 150 μL/well of Tris solution was added, and the solution was shaken at 37 °C for 20 min. The optical density value was measured at a wavelength of 540 nm under a microplate reader to calculate the cell proliferation activity.

### 2.8. UPLC Conditions and ESI-Q TRAP-MS/MS

The sample extracts were analyzed using a UPLC-ESI-MS/MS system and a C18 column (ACQUITY UPLC HSS T3 1.8 μm, 2.1 mm × 100 mm; Waters Corp., Milford, MA, USA). The mobile phase consisted of two solvents, A (water with 0.04% acetic acid) and B (acetonitrile with 0.04% acetic acid), and was run in a gradient elution as follows: 95:5 *v/v* (mobile phase A: mobile phase B) at 0 min, 5:95 *v/v* at 11.0 min, 5:95 *v/v* at 12.0 min, 95:5 *v/v* at 12.1 min, and 95:5 *v/v* at 15.0 min. The column temperature was set at 40 °C, and the injection volume was 2 μL. The flow rate was 0.40 mL/min. After UPLC, the effluent was alternatively connected to an ESI-triple quadrupole-linear ion trap mass spectrometer (Q TRAP-MS).

### 2.9. Qualitative and Quantitative Determination of Metabolites, Kyoto Encyclopaedia of Genes and Genomes (KEGG) Functional Annotation, and Enrichment Analysis

Based on the self-built database NWDB (Metware Biotechnology Co., Led. Wuhan, China) and the public database of metabolite information, the qualitative analysis of secondary mass-spectrometry data was performed. Each sample was analyzed twice, and data were captured in the positive and negative ion modes. In terms of the identification of metabolites, most peaks were identified by library searches with high matching scores. Multidimensional statistical analysis was used to establish a reliable mathematical model for the analysis of metabolites. The variables with variable weight values > 1 were considered as differential variables. In the permutation test, the sequential order of the categorical variable Y was randomly changed many times (*n* = 200), and the corresponding orthogonal partial least squares discrimination analysis (OPLS-DA) model was established. The metabolites with variable importance in the projection (VIP) > 1, log2FC ≥ 2, or log2FC ≤ 0.5 were considered as differential metabolites. Based on the differentially expressed metabolites, we compared the groups and conducted KEGG pathway analysis to investigate the metabolic reactions and metabolic pathways in each species.

### 2.10. Data Processing and Statistical Analysis

The data were centered using the mean of three parallel experiments. The results were analyzed by Statistical Analysis System 9.2 (SAS Institute Inc., Cary, NC, USA). *p* < 0.05 were considered statistically significant.

## 3. Results

### 3.1. Antioxidant Activities and α-Glucosidase Inhibition Activity of WS and CS

The results of DPPH radical scavenging activities of WS and CS are shown in Figure 1A. It showed a concentration-dependent manner, and the WS extracts showed higher scavenging activities compared to the CS samples, with an IC_50_ of 0.03 and 0.27 mg/mL, respectively. At the concentration of 1 mg/mL, the WS extract had a scavenging rate of 82.58%, while the CS extract had a scavenging rate of only 71.08% (*p* < 0.05). As shown in Figure 1B, the ABTS scavenging activities of the two extracts exhibited a concentration-dependent manner. The scavenging effects of WS samples on ABTS radicals were greater than that of CS samples at the same concentration, with an IC_50_ of 0.77 and 2.12 mg/mL, respectively. At the concentration of 1 mg/mL, the WS extract had a scavenging rate of 84.28%, while the CS extract had a scavenging rate of only 38.01% (*p* < 0.05).

The two extracts exhibited a dose-dependent effect (Figure 1C.). The extracts from WS (84.82 ± 0.34%) showed better α-glucosidase inhibitory activity than those from CS (56.06 ± 0.24%) (*p* < 0.05). The inhibitory effects of the two extracts were lower than that of acarbose. After stimulation with H_2_O_2_, the cell proliferation inhibition rate reached 90%, with shrinkage, irregular shapes, and lower adherence observed in the cells (Figure 1D). However, after pre-incubation with CS extracts or WS extracts, the inhibition rate was reduced to 73.94 ± 3.06% and 80.44 ± 2.77%, respectively, which were significantly lower (*p* < 0.05) than the group treated with H_2_O_2_ (90.69 ± 4.39%). The group pre-incubated with WS extract showed no significant differences compared with the group pre-incubated with acetylcysteine. Additionally, the cell morphology tended to be normal in the group treated with CS extract and WS extract.

### 3.2. Difference in Total Flavonoid and Total Phenolic Contents between WS and CS

TFC and TPC differences between WS and CS were determined. As shown in Figure 1E,F, the results showed that TFC and TPC levels in WS were 219.51 ± 5.18 mg QE/g and 41.53 ± 1.25 mg GAE/g, respectively, which were significantly higher than those in CS (*p* < 0.05).

### 3.3. Non-Targeted Analysis of Metabolites by UPLC-Q-TOF-MS/MS

An overlay of a typical total ion current chromatogram plot of quality control (QC) samples by mass spectrometry detection and the multi-peak detection plot of metabolites from multiple reaction monitoring (MRM) in negative and positive ion modes is shown in Figure 2. The plots depict a continuous mapping of the intensity addition of ions in the mass spectrogram at different time points. The high degree of overlap indicates good repeatability and instrumental stability during the detection at different times, which indicates that the methods had high reliability. In this study, 642 metabolites belonging to various phytochemical classes were identified, comprising 87 amino acid derivatives, 41 phenylpropanoids, 14 alcohols, 16 polyphenols, 12 phenolamines, 51 nicotinic acid derivatives, 16 anthocyanins, 68 flavones, 24 flavonols, 22 flavonoids, 16 flavonones, 39 alkaloids, 21 carbohydrates, 8 terpenoids, 16 vitamins and their derivatives, 14 isoflavones, 9 indole derivatives, 77 organic acids, 4 proanthocyanidins, 5 steroids, 58 lipids, and 24 other metabolites. The flavonoid metabolites identified are shown in Appendix A.

### 3.4. Differential Metabolite Analysis Based on Principal Component Analysis (PCA)

The PCA score scatter plots for all samples including the QC samples are shown in Figure 3A. The distinction and the QC group was significant based on the top-ranking principal components (PCs), indicated that the metabolites were significantly different between them. In the inter-group comparison, the 2D scatter plot of the scores on the first two PCs showed that PC1 and PC2 together accounted for 79.03% of the total variance (69.03% and 10%, respectively). The results suggested that WS could be clearly separated and identified on the basis of the metabolic profiles. The distinct separation indicated a large difference in metabolic phenotypes between the two soybean species.

### 3.5. Differential Metabolite Analysis Based on OPLS-DA

OPLS-DA was conducted as a supervised method to investigate the differences in metabolites with better discrimination. The score scatter plot inferred from the inter-group comparison of the two soybean species is shown in Figure 3B. The Q2 value was greater than 0.9 and indicates that the model was robust. The values of R2Y and Q2 were 1 and 0.983, respectively, which demonstrated that these models were stable and reliable and could be used to further screen metabolite differences.

The screening results based on VIP are shown in Appendix A. The comparison of metabolites between the two species identified 238 differential metabolites (*p* < 0.05, VIP > 1), with 151 upregulated and 87 downregulated (Figure 4C). For a better overview of the differential metabolites, a heat map of six samples versus 238 metabolites was plotted (Figure 3C). The hierarchical cluster analysis enabled the classification of metabolites with the same characteristics and identification of inter-group variation in metabolite characteristics. A clear grouping pattern could be observed by comparing the color intensity across all samples.

### 3.6. Identification of Differential Metabolites between WS and CS

To screen the best discriminant variables for classifying the species, the metabolites were ranked by fold change. The differential metabolites were classified into 20 categories (Figure 4B): 79 flavonoids (including 6 phenolic acids), 27 flavones, 9 flavonols, 11 flavonoids, 10 flavanones, 7 isoflavones, 6 anthocyanins, 3 procyanidins, 20 amino acids and derivatives, 39 lipids, 27 organic acids and derivatives, 15 phenylpropanoids, 13 nucleotides and derivatives, 14 alkaloids, 6 saccharides, 6 vitamins and derivatives, 4 phenolamines, 2 indoles and derivatives, 1 terpene, and 6 others. The top 20 significantly different compounds are shown in Figure 4D, including organic acids, isoflavones, flavanones, phenylpropanoids, alkaloids, and amino acids. The top 10 compounds that had significantly higher levels in WS were citric acid monohydrate, 2’-hydroxygenistein, garbanzol, octadecadien-6-ynoic acid, ligustilide, deoxyguanosine, protocatechuic aldehyde, ethyl cinnamate, 3-hydroxyanthranilic acid, and resveratrol. The CS samples contained significantly higher amounts of 2-furanoic acid, scoparone, 5,7-dihydroxy-3’,4’,5’-trimethoxyflavone, glucono-1,5-lactone, mandelic acid, skimmin, L-saccharopine, L-saccharopine, 3,7-di-O-methylquercetin, 4-phenyldiazenylbenzene-1,3-diamine, hydrochloride, and C-hexosyl-apigenin C-pentoside.

### 3.7. Identification of Differential Flavonoids Metabolites between WS and CS

From the 79 flavonoid compounds, 42 were significantly upregulated in WS, while 37 were significantly upregulated in CS. As shown in Figure 5, several subclasses of flavonoids (including isoflavones, flavones, flavonoid, flavonols, flavanones, proanthocyanidins, and anthocyanins) putatively annotated in the present study were significantly different between WS and CS.

With respect to the content of flavanones, the WS samples presented seven characteristic compounds, with garbanzol showing a significantly higher fold change than that in the CS samples (Figure 5A). The flavones detected in WS and CS, which included 27 compounds, are shown in Figure 5B; 12 compounds had significantly higher levels in WS, while 17 compounds had higher levels in CS, with apigenin being the glucoside with the highest content. The results indicated that the accumulation of flavones, including luteolin 7-*O*-glucoside, butin, apigenin, and narirutin, was greater in WS. Among the flavonoid compounds, engeletin, morin, and isovitexin 7-*O*-glucoside had significantly higher levels in WS, while 5,7-dihydroxy-3,4,5-trimethoxyflavone had a higher level in CS. In addition, eight flavonoid compounds had a significantly higher content in CS than those in WS (Figure 5C). 3,7-Di-*O*-methylquercetin, a flavonol, had a higher content in CS, while that of di-O-methylquercetin was higher in WS (Figure 5D). All the isoflavones presented in this study, including 2’-hydroxygenistein, genistein, sissotrin, glycitein, calycosin, glycitin, and genistein 7-*O*-glycoside, had significantly higher levels in WS than in CS (Figure 5E) and were useful in discriminating between WS and CS. In the polyphenol class, protocatechuic aldehyde and protocatechuic acid had significantly higher fold changes in WS than those in CS (Figure 5F).

### 3.8. KEGG Annotation and Metabolic Pathway Analysis of Differential Metabolites in WS and CS

The differential metabolites in WS and CS were functionally annotated using the KEGG database. We sorted out all the pathways to map the differential metabolites using KEGG classification, and pathways that accounted for percentages above 10% are shown in Figure 6A. The top KEGG pathways involved metabolism, biosynthesis of secondary metabolites, and flavonoid, flavone, and flavonol biosynthesis. Detailed KEGG pathway diagrams for WS versus CS are shown in Figure 6B, and the pathway analysis of the differential metabolites is presented in a bubble plot. In general, pathways are annotated as to their importance on the basis of enrichment analysis. The metabolic pathways related to the biosynthesis of flavonoids and phenolic acids, ranked using enrichment analysis in terms of the number of metabolites involved, are presented in Appendix A and include flavonoid biosynthesis, isoflavonoid biosynthesis, biosynthesis of phenylpropanoids, and flavone and flavonol biosynthesis (*p* < 0.05 in the enrichment analysis).

## 4. Discussion

In this study, the TPC, TFC, radical scavenging activities against DPPH and ABTS, and α-glucosidase values for WS and CS were compared. All values were significantly higher in WS than in CS. The TFC and TPC levels in WS were 8.2 and 3.4 times higher, respectively, than in CS. The DPPH and ABTS assay values for WS were 1.2 and 2.2 times higher, respectively, than for CS at 1 mg/mL. The α-glucosidase activity of WS was 1.5 times higher than that of CS. The results are consistent with those of previous studies, showing that, compared with small CS, large WS contained a higher amount of antioxidant molecules such as flavonoids and other polyphenolic components [15]. Similarly, the antioxidant activity and the phenolic compound content of small grain rice were higher than that of large grain rice [16]. Thus, breeders should pay attention to the cultivation of small grain varieties when breeding high phenolic content varieties.

In biological systems, oxidants and antioxidants have different chemical and physical characteristics. Therefore, no single assay can truly represent total antioxidant activity in a system. A battery of tests to measure free radical scavenging activity and antioxidant potential in the extracts is recommended [17]. Consequently, in the present study, a number of test systems were applied to measure free radical scavenging and antioxidant activity presented in both WS and CS extracts. The antioxidant activity of the WS and CS extracts was determined as the ability to scavenge free radicals (ABTS + and DPPH). The DPPH assay is widely used to evaluate the antioxidant ability of natural extracts. When the antioxidants in an extract donate electrons or hydrogen ions to DPPH, the mixture solution noticeably changes from purple to yellow [18]. The ABTS assay, which is based on electron transfer reactions that convert radicals to stable products, is used to measure the total antioxidant activity of a potential antioxidant [19]. Our results are consistent with those of previous studies, in which both ABTS and DPPH assays revealed higher antioxidant activities in WS than in CS [20]. Moreover, WS extracts presented better protective effects against H_2_O_2_-induced oxidative injury in a human endothelial cell line than CS extracts. The results from the present study also supported the results obtained by the ABTS and DPPH assays. Digestive enzymes, such as α-glucosidase, are key enzymes that can break down carbohydrates into glucose before uptake into the blood [21]. In this test, the two extracts obtained from CS and WS demonstrated inhibitory activity against α-glucosidase, and WS extracts exhibited better activity than CS extracts did, which indicated that WS can be used as a natural food resource in the treatment of diabetes.

In our previous study, WS was confirmed to have better health benefits compared to CS [4]. Previous studies have reported that the beneficial health effects of black soybean are due to its secondary metabolites, especially phenols and flavonoids [22]. In this study, an untargeted metabolomic method was applied to analyze phytochemical metabolite composition, mainly focusing on flavonoid compounds.

PCA was initially applied to better visualize the data by reducing multivariate data to a two-dimensional plot and presenting overall sample clustering [23]. It was conducted by automated modeling analysis after the data were log-transformed and mean-centered. In order to evaluate the metabolic profile differences between WS and CS, the data of 642 metabolites were subjected to PCA to discriminate the variations between the two species. In the inter-group comparison, PCA results indicated significant differences in metabolites between WS and CS. In contrast, there were no significant differences between the samples in the sample group, indicating the uniformity of composition within a soybean species. OPLS-DA has better classification efficiency than PCA does because it filters system noise and extracts variable information [24]. To systematically understand the metabolic variations between WS and CS, the changes in significantly different metabolites were evaluated by calculating the fold-change and VIP values from the OPLS-DA model.

In a previous study, Hyeon, Xu, Kim, and Choi reported the differentiation of distinctive leaf metabolites between cultivated and semiwild soybean using a metabolomics approach and found marked differences in flavonoids, and only three secondary metabolites were detected [7]. Natarajan, Xu, Bae, Caperna, and Garrett employed a proteomic approach to separate, identify, and compare storage proteins in wild and cultivated soybean [25]. A few studies focused on secondary metabolites, and especially flavonoids, in WS and CS. To our knowledge, this is the first untargeted metabolomic study identifying differences in flavonoids between WS and CS.

From the 238 metabolites identified in this study, the top 10 compounds with VIP > 1.2 in WS were selected as marker compounds. Among the top 10 significantly different compounds in WS, seven are well known to exhibit excellent health benefits. These include three phenolic compounds (2-hydroxygenistein, garbanol, and protocatechuic aldehyde), two phenylpropanoids (ethyl cinnamate and resveratrol), one organic acid and its derivatives, one lipid, and ligustilide. Flavone and isoflavone derivatives have drawn much attention because of their antioxidant radical scavenging capacity, with isoflavone showing better antioxidant capacity than flavone [26]. In the present study, the isoflavone compound 2-hydroxygenistein had a significantly higher level in WS than in CS. Protocatechuic aldehyde has potent antioxidant and antitumor activity and suppresses cisplatin-induced injury of tubular epithelial cells [27]. Ligustilide has been reported to have various biological activities, such as smooth muscle relaxation, antiplatelet aggregation, and analgesia. Resveratrol has been shown to have anti-inflammatory and antioxidant activity and affects the initiation and progression of many diseases [28]. Ethyl cinnamate, a cinnamic acid ester, has already been approved as a flavor additive by the US Food and Drug Administration and has proven antifungal properties [29].

Among the top 10 significantly different compounds in CS, several exhibited excellent activities. Scoparone has antioxidative and hepatoprotective effects and can regulate lipid metabolism in animals [30]. Skimmin can suppress streptozotocin-induced diabetic nephropathy, has anti-inflammatory activity, and inhibits immune complex deposition [31]. Therefore, the phytochemical analysis suggested that the differential metabolites in the two soybean species may contribute to their functional differences in human health.

Flavonoids are important bioactive components in soybean and are synthesized by the phenylpropanoid metabolic pathway, which yields flavanones, dihydroflavonols, and, subsequently, anthocyanins. Many flavonoid compounds can be formed via the flavonoid biosynthesis and isoflavonoid biosynthesis metabolic pathways [32]. In the current study, 79 differentially accumulated flavonoids were identified in WS and CS, and flavones in particular were detected in abundance (27 types). The secondary metabolites including phenolic and flavonoid compounds in edible plants have gained attention due to their remarkable biological activity [33]. 2′-Hydroxygenistein, which in this study had the highest fold-change value among compounds in CS, has been reported to suppress the binding of [3H]-dihydrotestosterone to its estrogen and androgen receptors and possesses a greater antioxidant capacity than other forms of hydroxygenistein [26]. Apigenin has been used in humans to treat a number of disorders and inflammatory conditions [34]. Previous studies have reported that three flavonoids, epicatechin, procyanidin B2, and cyanidin-3-O-glucoside, were the major antioxidant compounds in WS [7], but in this study, the three compounds all had higher levels in CS. The differences may be due to the different geographical origins. Future studies should identify the candidate genes for flavonoid biosynthesis in WS and study the functions of the flavonoid biosynthesis genes.

## 5. Conclusions

The total phenolic and flavonoid content was significantly higher in WS than in CS. A battery of tests on antioxidants indicated that the extracts from WS exhibited greater free radical scavenging abilities against DPPH and ABTS and better protective effects against H_2_O_2_-induced oxidative injury in HUVECs than those from CS. Moreover, a UPLC-QqQ-MS-based metabolomic approach was used to evaluate the differences in flavonoid compounds between WS and CS. In the present study, 79 flavonoid compounds were identified, and 42 were significantly upregulated in WS, while 37 were significantly upregulated in CS. The top four metabolic pathways, ranked using enrichment analysis in terms of their numbers of metabolites, involved the biosynthesis of secondary metabolites and the biosynthesis of flavonoids, flavones, and flavonols. This study substantially elucidated differences in the content of flavonoids and the biological activities between WS and CS, which is useful information for the effective utilization of these two black soybean species in food processing.

## Figures and Tables

**Figure 1 plants-10-00813-f001:**
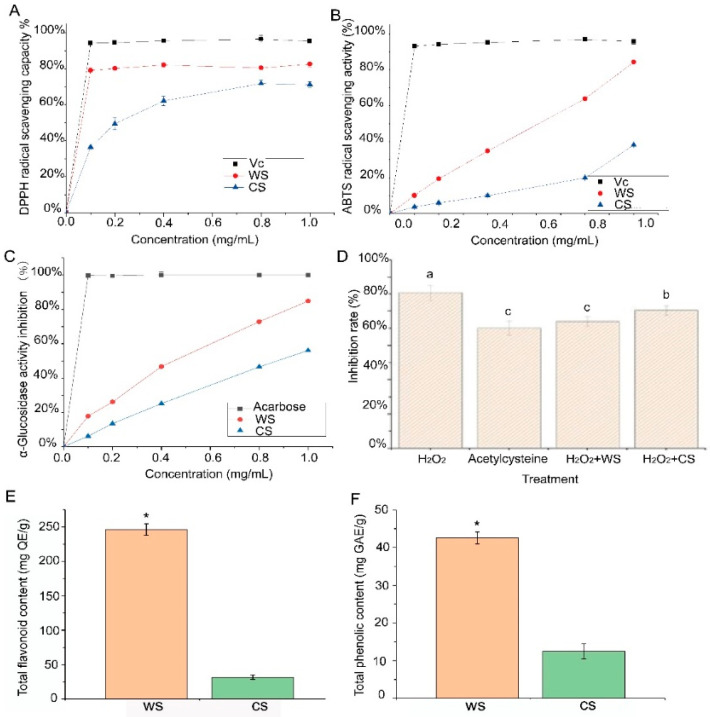
Biological activity and total phenolic and flavonoid content of wild soybean (WS) and cultivated soybean (CS). (**A**) DPPH radical scavenging activity; (**B**) ABTS radical scavenging activity; (**C**) α-glucosidase inhibitory activities; (**D**) inhibition rate after pre-incubation with WS and CS; (**E**) total flavonoid content of WS and CS; (**F**) total phenolic content of WS and CS. Each value represents means ± SD of triplicates. Statistical significance with an asterisks is according to *p* < 0.05.

**Figure 2 plants-10-00813-f002:**
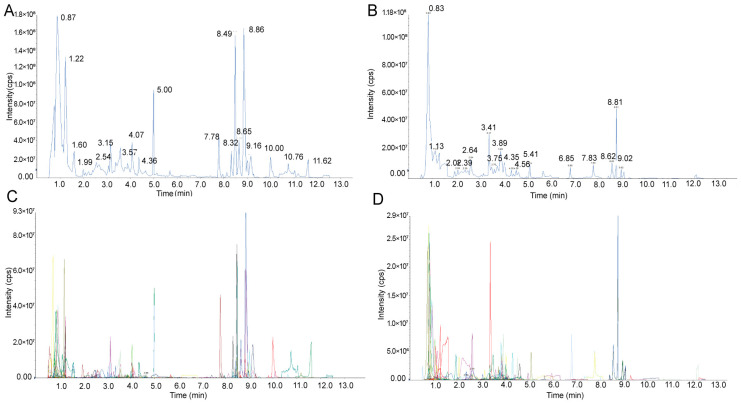
Total ion current of one quality control sample by mass spectrometry detection in (**A**) positive mode and (**B**) negative mode. Multi-peak detection plot of metabolites from multiple reaction monitoring in (**C**) positive mode and (**D**) negative mode.

**Figure 3 plants-10-00813-f003:**
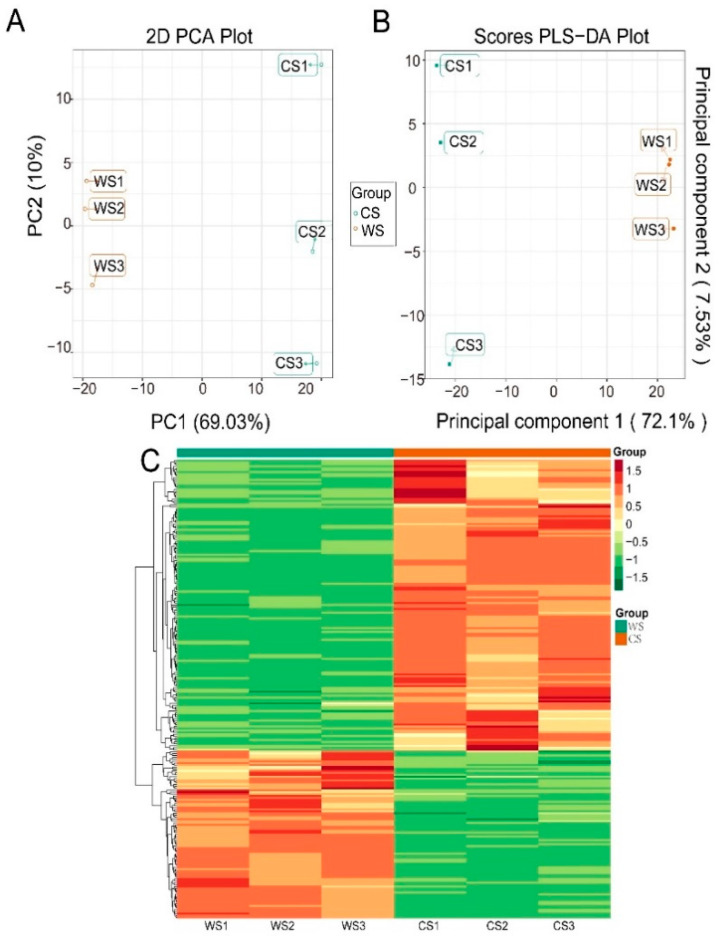
PCA and OPLS-DA results for the WS versus CS groups. PCA scatter plots: (**A**) principal component (PC_1_ versus PC_2_) (2D); (**B**) score scatter plot of the OPLS-DA model; (**C**) clustering heat map of the differential metabolites in WS and CS. The metabolite content data were normalized using maximum difference normalization. Each metabolite is represented by a row and each sample by a column. The color represents the abundance of each metabolite.

**Figure 4 plants-10-00813-f004:**
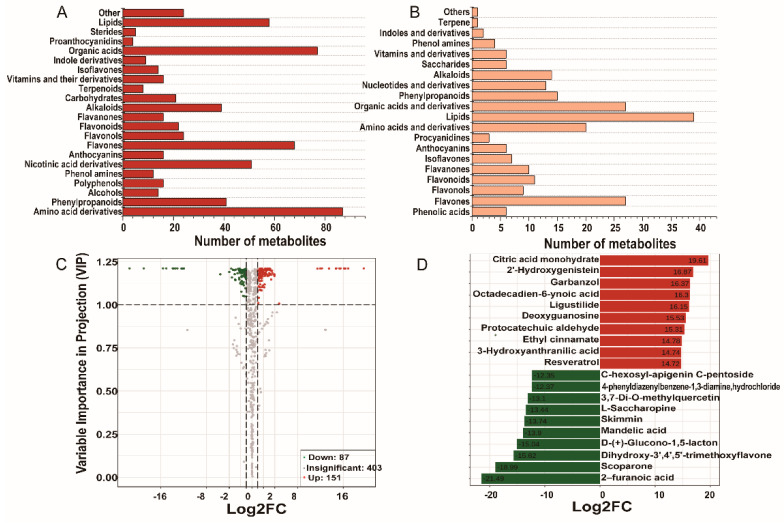
Different metabolites in WS and CS. (**A**) Number of different types of total metabolites. (**B**) Number of different types of significantly different metabolites. (**C**) Volcano plot. Significantly up- and downregulated metabolites are indicated in red and green, respectively. Those without significant difference between the two groups are indicated in grey. A greater absolute value on the horizontal axis indicates a greater change between WS and CS. A greater value on the vertical axis indicates greater significance of the relative differential content. (**D**) Histogram of the fold changes of the metabolites of the first 20 compounds that differed between WS and CS. The abscissa represents the log2 of multiples, and ordinates represent the different metabolite codes.

**Figure 5 plants-10-00813-f005:**
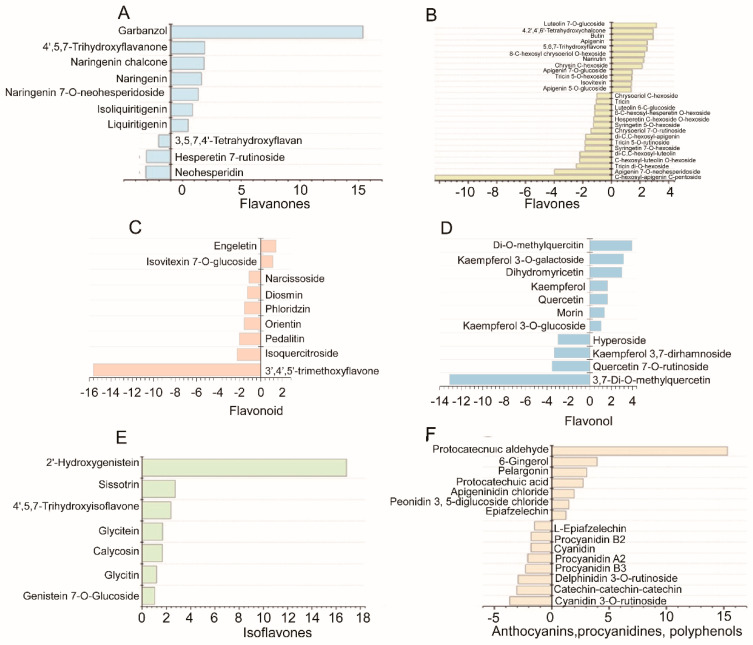
Significant differential flavonoids metabolites in WS and CS. (**A**–**F**) indicate the log2-fold-change of flavanones, flavones, flavonoid, flavonol, isoflavones and anthocyanins, procyanidines polyphenol, respectively, in WS versus CS.

**Figure 6 plants-10-00813-f006:**
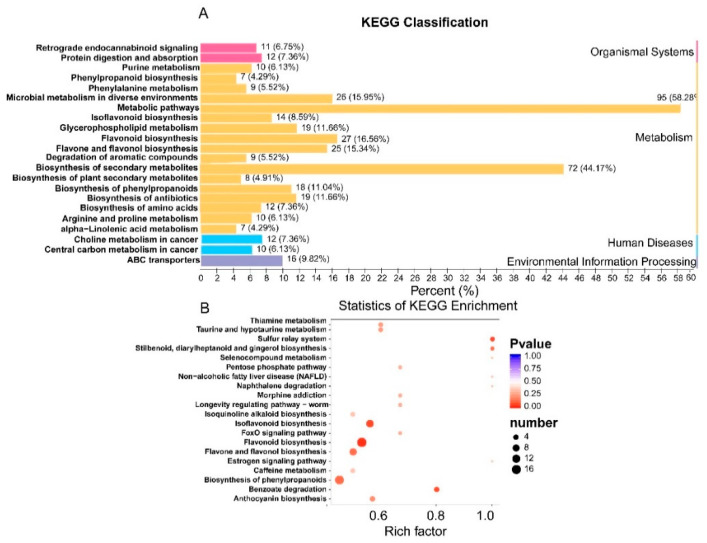
KEGG classification and enrichment of differential flavonoid metabolites for WS versus CS. (**A**) The vertical axis represents the names of the KEGG metabolic pathways, while the horizontal axis gives the ratio of metabolites annotated to a pathway to the total number of annotated metabolites in percentage values. (**B**) The vertical axis represents the enrichment factor of each pathway, while the horizontal axis shows the names of the pathways. The color of each point in this figure represents its *P*-value; the redder a point, the more significant its enrichment. The size of a point represents the number of differential metabolites that have been annotated to its corresponding pathway.

## Data Availability

Research data available through the corresponding author.

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
