# Peer review of "Comparison of Phenolic and Flavonoid Compound Profiles and Antioxidant and α-Glucosidase Inhibition Properties of Cultivated Soybean (Glycine max) and Wild Soybean (Glycine soja)"

_plants, 2021, doi:10.3390/plants10040813_

Round 1
Reviewer 1 Report
The abstract must be completely revised for the language
introduction line 43: revise “and normally be used”
line 56: change to “involves”
line 68: change to “flavonoids and phenolics”
paragraph 2.2: Pancreatic lipase is not mentioned in the text regarding the results
line 89: please italicize “Saccharomyces cerevisiae”
paragraph 2.6: why the authors selected this cell line?
line 147: dyed?
paragraph 3.1: for DPPH and ABTS assays the authors must provide the IC50 values for the extracts and reference compounds
Regarding HUVEC cell line, the readers must be see the different morphology with or without treatment and know the viability after treatment with the extracts without the pretreatment with hydrogen peroxide
Figure 1 is not clear. The legend must be implemented with statistics and explanation of the acronyms. Vc? asterisks?
line 438: pay attention to the ref 34
Reviewer 2 Report
In the presented manuscript, the authors compared total phenolic and flavonoid contents, antioxidant activity, α-glucosidase inhibition activities and protective effects against H2O2-induced oxidative injury in a human endothelial cell line of the wild and cultivated soybean. Moreover, they used metabolomics as a tool to evaluate the specific flavonoid and polyphenol composition in wild soybean for the first time. The work is essential and comprehensive. The manuscript is overall written very well. However, I have a few comments.
The extraction of phenolic compounds and α-Glucosidase inhibitor activity assay should be described in brief for completeness.
Line 143: H2O2, the numbers should be written in subscript.
Lines 152 and 216: UHPLC or UPLC?
Reviewer 4 Report
The English language should be improved, since the whole text contains unclear phrases:
“extracts exhibited increased levels of inhibitory activity with increasing” – do the authors mean a dose-dependent effect? (line 193).
“higher fold changes in CS than those in WS” – does it mean higher content? (e.g. lines 310, 311).
The use of subscripts and superscripts should be checked (e.g. lines 142, 143 and elsewhere).
Many abbreviations need explanation: MRM, PCA, OPLS-DA, VIP, R2Y, etc.
Why the authors use the term flavonoids next to flavonols, isoflavones, etc? (lines 226, 270, 328, Fig. 5). Flavonols, isoflavones and so on are different classes of flavonoids.
Introduction.
Line 71: What “functional components” were identified?
Materials and methods.
Section 2.1: The samples should be described better, including the number of plants, part of the plant (bean?) should be pointed out, etc.
Section 2.3: Preparation of the extracts needs to be described properly. What amount of the material was extracted? Was the material milled before extraction? Solvent should be specified. How can we discriminate free and bound phenolics?
DPPH and ABTS radical scavenging asays are not biological activities (line 110).
Methods / Results
The authors should clearly specify in Methods or in Figure legends how the differences in the chemical composition were evaluated? Is it based on peak area?
Figure legends are unsatisfactory - concentrations used and incubation times should be specified (e.g. section 2.6 or respective Figure/result section).
Results.
Fig. 1D: Inhibition rate? What is it? Inhibition of cell growth? What is the concentration used?
Fig. 4: What does Log2FC mean? (also line 173)
Fig. 4D: Is Hydrochloride correct?
Fig. 5B is illegible.
Discussion.
This part could be shortened.
We can read that “ABTS/DPPH assays revealed higher activities in CS” (line 371). Is it really in agreement with the present study?
Round 2
Reviewer 1 Report
The authors have addressed all my comments and the manuscript can be accepted in this revised version.
Author Response
Dear,
Thank you very much for your help in reviewing of our manuscript. Your questions and suggestions are helpful for us to improve our manuscript.
Best regards.
Reviewer 4 Report
I recommend revision of Figures 4 and 5.
1) Figure 4D shows compound “hydrochloride”, but hydrochlorides are generally salts of organic compounds with HCl. It seems that the compound is not specified correctly.
2) Figure 5C shows flavonoids, but for example there are morin (a flavonol) and hyperoside (a flavonol glycoside). It is unclear why they are not shown together with other flavonols in Fig 5D.
